# Imatinib Optimized Therapy Improves Major Molecular Response Rates in Patients with Chronic Myeloid Leukemia

**DOI:** 10.3390/pharmaceutics14081676

**Published:** 2022-08-12

**Authors:** Hyacinthe Johnson-Ansah, Benjamin Maneglier, Françoise Huguet, Laurence Legros, Martine Escoffre-Barbe, Martine Gardembas, Pascale Cony-Makhoul, Valérie Coiteux, Laurent Sutton, Wajed Abarah, Camille Pouaty, Jean-Michel Pignon, Bachra Choufi, Sorin Visanica, Bénédicte Deau, Laure Morisset, Emilie Cayssials, Mathieu Molimard, Stéphane Bouchet, François-Xavier Mahon, Franck Nicolini, Philippe Aegerter, Jean-Michel Cayuela, Marc Delord, Heriberto Bruzzoni-Giovanelli, Philippe Rousselot

**Affiliations:** 1Department of Hematology, CHU Côte de Nacre, 14000 Caen, France; 2Pharmacology Department, Centre Hospitalier de Versailles, 78150 Le Chesnay, France; 3Department of Hematology, CHU, Institut Universitaire du Cancer—Oncopole, 31100 Toulouse, France; 4Department of Hematology, Hôpital Bicêtre AP-HP, 94270 Bicêtre, France; 5Department of Hematology, Centre Hospitalier Pontchaillou, 35000 Rennes, France; 6Department of Hematology, CHU d’Angers, 49100 Angers, France; 7Department of Hematology, Centre Hospitalier Annecy Genevois, 74370 Pringy, France; 8Department of Hematology, Hôpital Huriez—CHRU, 59000 Lille, France; 9Department of Hematology, Hôpital Victor Dupouy, 95107 Argenteuil, France; 10Department of Hematology, Hôpital de Meaux, 77100 Meaux, France; 11Department of Hematology, Centre Hospitalier de Dieppe, 76202 Dieppe, France; 12Department of Hematology, Hôpital de Dunkerque, 59240 Dunkerque, France; 13Department of Hematology, Hôpital de Boulogne, 62200 Boulogne, France; 14Department of Hematology, Hôpital Notre Dame de Bon Secours, 57000 Metz, France; 15Department of Hematology, Hôpital Cochin APHP, 75014 Paris, France; 16Research Department, Centre Hospitalier de Versailles, 78150 Le Chesnay, France; 17Inserm CIC 802, CHU de Poitiers, 86000 Poitiers, France; 18Department of Pharmacology, Centre Hospitalier Pellegrin—Tripode, 33000 Bordeaux, France; 19Department of Hematology, Institut Bergonié, 33076 Bordeaux, France; 20Department of Hematology, Centre Léon Bérard, 69008 Lyon, France; 21Unité de Recherche Clinique et Département de Santé Publique, GIRCI Ile de France, Hôpital Ambroise Paré, 92100 Boulogne, France; 22Hematology and Molecular Biology and EA3518, Hôpital Saint-Louis, AP-HP, 75010 Paris, France; 23Hôpital Saint-Louis, AP-HP/INSERM, Université Paris-Cité et CIC 1427, 75010 Paris, France; 24Department of Hematology, Centre Hospitalier de Versailles, 78157 Le Chesnay, France; 25UMR1184, IDMIT Department, Commissariat à L’énergie Atomique et aux Energies Alternatives, University of Versailles Saint-Quentin-en-Yvelines Paris-Saclay, 92265 Fontenay-Aux-Roses, France

**Keywords:** imatinib, therapeutic drug monitoring, chronic myelogenous leukemia

## Abstract

The registered dose for imatinib is 400 mg/d, despite high inter-patient variability in imatinib plasmatic exposure. Therapeutic drug monitoring (TDM) is routinely used to maximize a drug’s efficacy or tolerance. We decided to conduct a prospective randomized trial (OPTIM-imatinib trial) to assess the value of TDM in patients with chronic phase chronic myelogenous treated with imatinib as first-line therapy (NCT02896842). Eligible patients started imatinib at 400 mg daily, followed by imatinib [C]min assessment. Patients considered underdosed ([C]min < 1000 ng/mL) were randomized in a dose-increase strategy aiming to reach the threshold of 1000 ng/mL (TDM arm) versus standard imatinib management (control arm). Patients with [C]min levels ≥ 1000 ng/mL were treated following current European Leukemia Net recommendations (observational arm). The primary endpoint was the rate of major molecular response (MMR, BCR::ABL1^IS^ ≤ 0.1%) at 12 months. Out of 133 evaluable patients on imatinib 400 mg daily, 86 patients had a [C]min < 1000 ng/mL and were randomized. The TDM strategy resulted in a significant increase in [C]min values with a mean imatinib daily dose of 603 mg daily. Patients included in the TDM arm had a 12-month MMR rate of 67% (95% CI, 51–81) compared to 39% (95% CI, 24–55) for the control arm (*p* = 0.017). This early advantage persisted over the 3-year study period, in which we considered imatinib cessation as a censoring event. Imatinib TDM was feasible and significantly improved the 12-month MMR rate. This early advantage may be beneficial for patients without easy access to second-line TKIs.

## 1. Introduction

Chronic myeloid leukemia (CML) is a myeloproliferative disorder associated with the t(9;22)(q34;q11.2) translocation and its cytogenetic hallmark, the Philadelphia chromosome (der22). This translocation results in a *BCR::ABL1* fusion gene that codes for a BCR::ABL1 oncoprotein (p210 BCR::ABL1), which causes enhanced and deregulated tyrosine kinase activity [1]. The ability of BCR::ABL1 to induce a similar disease in mice pushed the design of tyrosine kinase inhibitors (TKIs), a new class of anticancer agents led by imatinib [2,3,4,5]. The TKIs class is now enriched with second-generation compounds (dasatinib [6], nilotinib [7] and bosutinib [8]) registered as first and second-line therapies. Ponatinib [9], a third generation TKI, has been registered in subsequent lines or in case of T315I mutation.

Imatinib is approved in chronic phase CML (CP-CML) at the dose of 400 mg once daily. Second-generation TKIs have been compared against the standard imatinib dose in first-line chronic phase CML, demonstrating faster kinetics as the molecular response without a survival advantage [10,11]. Thus, international recommendations for first-line CP-CML still include imatinib as a first-line therapeutic option [12,13,14,15]. The safety of long-term imatinib therapy is well established [5] compared to other front-line options, such as dasatinib 100 mg once a day (28% of patients will experience pleural effusions by 5 years [10]), nilotinib 300 mg twice a day (13% of the patients will develop cardiovascular events by 5 years [11]) or bosutinib 400 mg daily (7.8% diarrhea grade 3, 19% increased ALT [16]). The recent release of generic imatinib also raised the question of cost-effective strategies. For example, generic imatinib given as frontline therapy, followed, if necessary, by second-generation TKIs, has been shown to be a cost-effective strategy [17,18].

Imatinib dose optimization has been evaluated in several prospective clinical studies testing the use of high-dose imatinib (600 mg to 800 mg daily). A systematic review and meta-analysis of randomized controlled trials comparing frontline treatment with imatinib 400 mg daily versus high doses concluded that these strategies resulted in an increase in toxic effects with a minimal therapeutic advantage [19,20]. Pharmacokinetic studies pointed out the importance of inter-patient variability in imatinib plasma trough concentrations ([C]min), varying by 55 to 106% among patients under a given dosage [21]. Imatinib [C]min correlates with pharmacodynamic responses, and it has been suggested in a retrospective study that the threshold of 1000 ng/mL was associated with an improved molecular response in patients treated with imatinib 400 mg daily [22,23,24].

Therapeutic drug monitoring (TDM) is routinely considered for the management of medications to avoid or control adverse events and to maximize efficacy. We thus decided to initiate the randomized multicentric “OPtimized Tyrosine kInase Monotherapy for imatinib study (OPTIM-imatinib study)” in order to demonstrate, prospectively, the benefits of TDM based on the [C]min assessment in patients with CP-CML receiving imatinib as front-line treatment.

## 2. Methods

### 2.1. Patients and Synopsis of the Study Protocol

The OPTIM-imatinib study is a prospective, randomized, phase-2 trial conducted in centers of the French CML group (Fi-LMC). Adult CML patients were eligible if they were (i) newly diagnosed in the chronic phase for less than 13 weeks, not previously treated or treated with IM 400 mg daily for less than 13 weeks, (ii) not previously treated with tyrosine kinase inhibitors other than imatinib and (iii) provided signed, written inform consent. Women of childbearing potential had to use an adequate method of contraception. The study was registered as the OPTIM-imatinib trial, ClinicalTrials.gov NCT02896842. All patients gave their informed consent.

Imatinib [C]min was centrally determined by chromatography-tandem mass spectrometry 15 days after enrollment, as previously described [23]. Briefly, after a liquid–liquid extraction, imatinib and its deuterated internal standard were eluted on an XTerra RP18 column with a gradient of acetonitrile–ammonium formiate buffer 4 mmol/L, pH 3.2. Imatinib was detected by electrospray ionization mass spectrometry in multiple reaction-monitoring mode. The calibration curves were linear over the range 10–5000 ng/mL. The limit of quantification was set to 10 ng/mL. Patients with a [C]min < 1000 ng/mL were randomized among a dose-increase strategy aiming to reach the threshold of 1000 ng/mL (TDM arm) and standard imatinib management (control arm). Patients with [C]min levels ≥ 1000 ng/mL were observed (observational arm). Imatinib [C]min levels were assessed monthly in the TDM and control arms and every 3 months in the observational arm. All patients started therapy with imatinib 400 mg daily. In the absence of grade ≥ 2 adverse events, patients allocated to the TDM arm were told to increase the imatinib daily dose from 400 mg to 600 mg, and imatinib plasma dosage was remeasured. If the threshold of 1000 ng/mL was not achieved, then patients were told to increase the imatinib daily dose again from 600 to 800 mg. The maximum allowed imatinib dosage was 800 mg daily, and dosages such as 500 or 700 mg daily were accepted. All patients were managed according to the European Leukemia Net (ELN) 2009 recommendations amended with ELN 2013 and ELN 2020 recommendations) for efficacy and toxicity [13,14,15]. The minimum authorized imatinib dosage was 300 mg daily.

### 2.2. Response Definition and Primary Endpoint

The primary end-point was the percentage of patients achieving a major molecular response (MMR) at 12 months, as defined by BCR::ABL1/ABL1 ratio on the International Scale (IS) (BCR::ABL1^IS^) ≤ 0.1% according to the European Leukemia Net recommendations for minimal residual disease quantification [25,26]. Molecular assessments were performed in hospital laboratories of the “French quality control network for BCR::ABL1 quantification” (GBMHM, Groupe de Biologie Moléculaire des Hémopathies Malignes) and centrally validated in the reference laboratory for France (Dr JM Cayuela, Hôpital Saint-Louis, Paris, France). BCR::ABL1^IS^ levels were tested every 3 months over 12 months, and every six month until the end of the study [13,14].

### 2.3. Pharmacokinetic Analyses and Secondary Endpoints

Imatinib [C]min was determined by chromatography-tandem mass spectrometry as previously described [23]. Secondary endpoints included (i) safety and efficacy analyses at different time-points, (ii) relationship between plasma dose and efficacy or tolerance and (iii) progression-free survival, event free survival and overall survival. Follow-up data (36 months) were collected.

## 3. Statistics

Analysis was performed on an intent-to-treat basis. The baseline characteristics were compared by non-parametric tests: either the exact Fisher’s test for qualitative variables or the Kruskal–Wallis test for quantitative variables. Confidence intervals (CIs) were calculated at the 95% confidence level. Correlations between plasma concentrations of imatinib ([C]min) at steady state and imatinib daily dosage were assessed using linear regression. Censored endpoints (cumulative cytogenetic and molecular response rates and overall survival) and their associated 95% CI were estimated by the Kaplan–Meier method. Impacts of prognostic factors on censored endpoints were assessed using the Cox proportional hazard model. The proportional hazard assumptions were checked using the scaled Schoenfeld residual test.

The primary endpoint of this study was to analyze the rate of the major molecular response at 12 months in the TDM arm (trough plasma level < 1000 ng/mL with adapted therapy). The control arm (trough plasma level < 1000 ng/mL without adapted therapy) was the estimator of the reference rate. A sample size of 80 randomized subjects was calculated for TDM and control arms in order to test the null hypothesis of H0: *p* ≤ 0.25 and alternative H1: *p* ≥ 0.40 with one-sided type I error of 5% and 80% power. The observational arm (trough plasma level > 1000 ng/mL) was the estimator of the best expected response rate. No interim analysis was planned. Toxic effects were assessed continuously.

## 4. Results

### 4.1. Patients’ Characteristics

From September 2010 through March 2014, 139 CML patients were recruited and screened. In six patients, the initial [C]min was not assessed (three stopped imatinib before the dosage and three declined the dosage). Thus, 133 patients were studied. In 86 patients (64.6%), initial [C]min value was <1000 ng/mL (Figure 1, Table 1). These patients were randomized into the TDM arm (43 patients) and the control arm (43 patients). [C]min was ≥1000 ng/mL in 47 patients, and they were allocated to the observational arm. Median age at diagnosis was 64 years (27 to 87), and sex ratio (M/F) was 2.09. Sokal score was low and intermediate in 82% of patients. No differences in terms of age, sex ratio or Sokal risk score were observed between patients included in the TDM and control arms. However, the median age of the patients with high [C]min was significantly higher than that of patients with low [C]min (67 y versus 61 y, *p* = 0.007). Sixty-one patients (51%) started imatinib at 400 mg daily before being included in the study, as permitted by the inclusion criteria. Duration of imatinib before inclusion for these patients was 4 weeks. Clinical characteristics of the patients are depicted in Table 1.

### 4.2. Efficacy

Among the randomized patients with a [C]min < 1000 ng/mL, the rate of a major molecular response at 12 months (primary end-point) was 67% (95% CI, 51–81) for patients included in the TDM arm, and it was 39% (95% CI, 24–55) for patients monitored in the control arm (*p* = 0.017) (Figure 2A). The estimated cumulative incidence of MMR (CI-MMR) by 12 months was significantly higher in the TDM arm (74.4% (95% CI, 65.1–81.6)) compared to the control arm (48.8% (95% CI, 33–62.8)) (Figure 2B). The rate of a complete cytogenetic response at 12 months observed in at least one assessment was not different for the TDM strategy (81% (95% CI, 99–66)) compared to the control arm (74% (95% CI, 86–59)). For patients followed in the observational arm, i.e., patients with [C)min ≥ 1000 ng/mL, the rates of MMR at and by 12 months were 49% (95% CI, 34–64) and 53% (95% CI, 39–66), respectively.

Patients were followed for efficacy after the primary end-point assessment. The CI-MMR by 36 months without censoring the patients at imatinib cessation was not significantly different for patients randomized in the TDM arm (83.7% (95% CI, 77.2–88.4)) and for patients randomized in the control arm (74.4% (95% CI, 65.1–80.7)) (Figure 2B). The CI-MMR by 36 months was also calculated by censoring the patients at the time of imatinib cessation. In this analysis, a significant benefit was observed for patients included in the TDM arm (81.1% (95% CI, 73–86.9)) compared to patients randomized in the control arm (66.1% (95% CI, 52.7–76.5)) (Figure 2C). Cumulative incidence rates of MR4.5 by 36 months without censoring were 41% (95% CI, 23–58.2) for the TDM strategy, 40.8% (95% CI, 22.3–58.5) in the control arm and 35.7% (95% CI, 17.8–54) in the observation arm (*p* = 0.6).

### 4.3. Imatinib Administration

Imatinib daily dose was fixed at 400 mg for all patients at inclusion. Patients randomized to the TDM arm were subjected to an imatinib dose increase in the absence of grade ≥ 2 adverse events based on [C]min data. The results of the dose optimization strategy are summarized in Figure 3A. At month 1, 31 patients (72%) were able to increase theirs daily doses to the first dose level (600 mg), and among those, seven patients (22%) further increased their doses to the second step (800 mg) at month 3. Only one patient had to decrease the dose to 300 mg. At month 6, 21% of the patients receiving imatinib were treated at 400 mg, 42% at 600 mg (including 2 patients at 500 mg) and 32% at 800 mg daily. Two patients decreased their doses to 300 mg. Few dose adaptations occurred after month 6. At the end of the pharmacokinetic follow-up, none patients (21%) discontinued imatinib, and the proportions of patients treated at 300, 400, 500–600 and 800 mg daily were 6%, 21%, 38% and 35%, respectively. Overall, 73% of the patients received more than 400 mg imatinib daily. The resulting mean daily doses at inclusion, month 1, month 3, month 6, month 9 and month 12 were 400 mg, 544 mg (95% CI, 516–572), 567 mg (95% CI, 523–611), 587 mg (95% CI, 540–634), 591 mg (95% CI, 529–653) and 603 mg (95% CI, 543–663), respectively.

Only one patient included in the control arm increased the daily dose to 600 mg at month 3, whereas three patients reduced the daily dose to 300 mg, resulting in a mean daily dose of 398 mg (95% CI, 384–411). At 12 months, eight patients (19%) discontinued imatinib, and the proportions of patients treated at 300 and 400 mg daily were 9% and 91%, respectively. The mean daily dose was 391 mg (95% CI, 382–401) (Figure 3B). At 12 months, a similar rate of discontinuation was observed in the observational arm (15%, *p* = 0.75). In this group, 10 patients (33%) experienced a dose reduction to 300 mg, while the remaining 30 patients received 400 mg daily. The mean daily dose was 375 mg (95% CI, 361–389) (Figure 3C).

At 36 months, 79 patients (59.4%) were on imatinib, 24 (55.8%) in the TDM arm, 25 (58.1%) in the control arm and 30 (63.8%) in the observational arm. The mean daily doses were 518 mg (95% CI, 440–596), 395 mg (95% CI, 368–422) and 404 mg (95% CI, 357–450) in the TDM, control and observational arms, respectively (*p* = 0.008).

### 4.4. Imatinib [C]min Assessment

Imatinib was lower than the 1000 ng/mL threshold in 64.6% of patients. Table 2 shows significant improvement in the median [C]min after dose adaptation in the TDM arm (*p <* 0.0001) compared to standard management in the control and observational arms. The goal of increasing the median [C]min over 1000 ng/mL was achieved in the TDM arm. A non-significant decrease in [C]min was observed in the control and observational arms. In order to define the optimal threshold for imatinib [C]min, we conducted ROC analysis on data from the 90 patients included in the control and observational arms, and found that a [C]min of 1031 ng/mL at month 1 was related to the achievement of MMR at month 12.

### 4.5. Safety

Overall, 255 all grade adverse events (AE) (including recurrent AEs) were recorded during a 3-year period, 95 in the TDM arm, 69 in the control arm and 91 in the observational arm. Forty-seven (15%) were grade 3–4 AEs (15 in the TDM arm, 13 in the control arm and 19 in the observational arm). Appendix A recapitulates the distribution of AEs by category, grade and arm, excluding recurrent AEs. No unexpected AEs were recorded; two patients relapsed from a previously known cancer—one prostatic and one adenocarcinoma. A better tolerance profile was observed for patients included in the control arm (considered as under dosed by the [C]min criteria), compared to patients allocated to the observational arm or randomized in the TDM arm. Appendix A shows that hematological and skin toxicities were not related to [C]min at inclusion, whereas other symptoms, such as musculoskeletal and gastro-intestinal disorders, were equally present in patients from the TDM and observational arms, suggesting that TDM resulted in a translation from an under-dosed to a well-dosed toxicity profile. Ten patients (7.5%) died during the follow-up period, 7 from cancers, 1 from suicide, 1 from natural death and 1 from CML progression in the observational arm.

## 5. Discussion

The OPTIM-imatinib study is the first prospective randomized study testing imatinib TDM for patients diagnosed with CML in first-line treatment. The study met her primary objective. Our dose adaptation strategy resulted in a significantly higher rate of MMR at 12 months and higher CI-MMR rates in the TDM arm compared to the control arm. We also confirmed that TDM is feasible. At 12 months, median [C]min 971 ng/mL (95% CI; 830–1242) in the TDM arm was not different to that of patients who were not randomized (963 ng/mL (95% CI; 845–1098)) and significantly higher than that of the control arm (639 ng/mL (95% CI; 494–729)). The resulting mean daily dose of imatinib was, as expected, higher in the TDM arm (603 mg daily at 12 months) compared to 391 mg daily in the control arm. We also validated the well-recognized imatinib threshold of 1000 ng/mL for the achievement of MMR at month 12 (1031 ng/mL in our hands).

Despite these encouraging results, the cumulative incidence advantage observed in favor of the TDM arm during the first 12-month period was not significantly conserved by 36 months. This observation is not in line with previous studies comparing imatinib to second-generation TKIs: these studies reported a conserved advantage in terms of CI-MMR after the achievement of a faster response with the use of second-generation TKIs. In these analyses, patients were censored at the time of study treatment cessation [6,7,8]. We therefore analyzed our patients with systematic censoring in case of imatinib cessation and demonstrated a significant benefit in favor of the TDM strategy in the long run, suggesting that imatinib TDM may offer a benefit of a similar magnitude compared to a switch to second-generation TKIs [10,11,12].

Previous comparisons between imatinib 400 mg and imatinib 600–800 mg daily were conducted without the use of TDM. Two single-armed studies of IM800 observed higher MMR rates compared to historical controls [27,28]. Four randomized studies tried to demonstrate these observations. The TOPS study compared imatinib 800 mg/d to imatinib 400 mg daily in patients newly diagnosed with Sokal high-risk CP-CML. MMR was reached faster at 3 and 6 months with high-dose imatinib than with imatinib at 400 mg daily. MMR rates were similar between arms at 12 months [29]. A long-term follow-up of the TOPS study showed that MMR rates were identical at 42 months (51.6% vs. 50.2% for 400 and 800 mg/d, respectively), and there was not a survival advantage [30]. Similarly, the German CML IV study also reported higher 12-month MMR rates in patients included in the optimized high-dose imatinib group compared to the standard-dose imatinib group (59% versus 44%, *p* < 0.001); there was not a long-term advantage [31,32]. The French SPIRIT trial showed MMR rates at 12 months that were significantly higher for imatinib 600 mg vs. imatinib 400 mg [33]. No other differences were recorded at subsequent time points or in a longer follow-up [34]. The last study was a phase-2 randomized study conducted by the SWOG. The primary end-point (MR4 at 12 months) was achieved in favor of the imatinib 800 mg daily cohort after a follow-up of 12 months [35].

High-dose imatinib studies reported around 30% more toxicities with imatinib at 600–800 mg daily compared to imatinib at 400 mg/day. We also observed an increase in grade 1–2 toxicities but not in grade 3–4. Moreover, this increase was not observed for hematological or skin toxicities, whereas other symptoms, such as musculoskeletal and gastro-intestinal disorders, were more frequent when [C]min was around the 1000 ng/mL threshold, such as in the TDM arm and the observational arm. The DESTINY study reported that a dose reduction of imatinib translated to a better tolerance, which is in line with the toxicity profile of our patients having a [C]min < 1000 ng/mL, irrespective of the daily dose [36]. Permanent discontinuation due to toxicity or refusal was similar for all treatment arms.

In our study, the imatinib dose optimization strategy was dictated by a [C]min measurement performed 7–10 days after inclusion. Our 12-month pharmacological follow-up demonstrated that imatinib [C]min was stable over time in patients treated with imatinib at 400 mg daily. With a single [C]min assessment and based on the 1000 ng/mL threshold, we were able to define well dosed patients, who represented only 35.4% of our patient population. The median age of our patients (64 years) may suggest that this observation is applicable to a “real life” CML population. For the remaining patients eligible for a TDM strategy, one assessment allowed us to increase imatinib to 600 mg daily in 72% of them, whereas 22% increased the dosage to 800 mg daily after a second assessment.

In conclusion, only 1/3 of our patients on imatinib 400 mg daily were correctly dosed and may not have required imatinib dose escalation. Two-thirds of the patients were not correctly exposed to imatinib at the standard dose and may have benefited from individualized dose optimization using the TDM strategy. This tailored dose adaption strategy resulted in higher MMR rates at 12 months (67% vs. 39%), a magnitude in line with the results reported with second-generation TKIs. Our results strongly support the use of TDM to optimize and personalize the daily dose of IM front-line therapy.

## Figures and Tables

**Figure 1 pharmaceutics-14-01676-f001:**
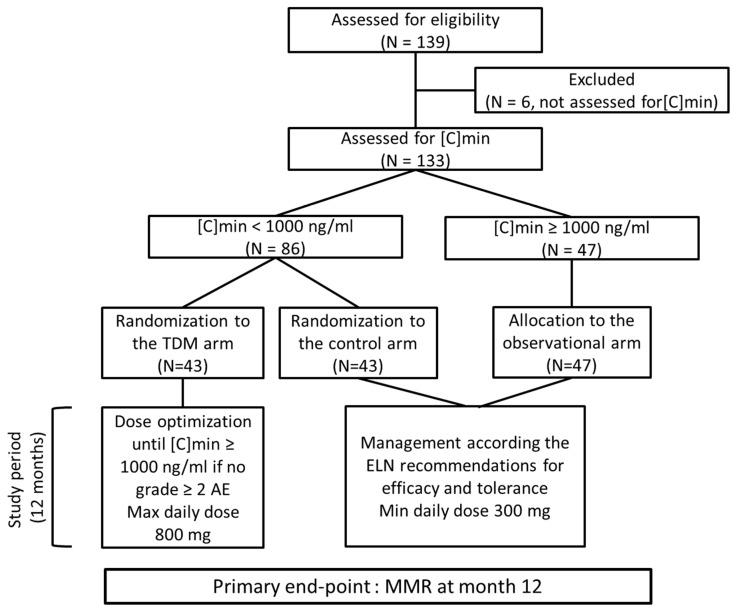
CONSORT diagram of the OPTIM-imatinib study. [C]min: trough imatinib plasma level, ELN: European Leukemia Net, AE: adverse event.

**Figure 2 pharmaceutics-14-01676-f002:**
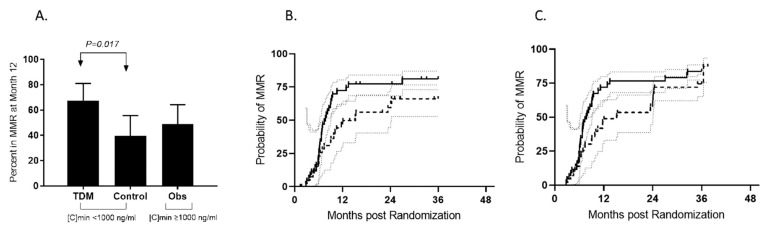
Major molecular response rates at 12 months (**A**) and by 36 months (**B**,**C**). (**A**) Twenty-nine patients (67% (95% CI, 51–81)) achieved MMR at month 12 in the TDM arm, as opposed to 39% (95% CI, 24–55) in the control arm (*p* = 0.017) (dark plots). The rate of MMR was 49% (95% CI, 34–64) for patients included in the observational arm (dark plots). TDM: therapeutic drug monitoring arm, MMR: major molecular response (dark plots), no MMR: grey plots, [C)min: trough imatinib plasma level at inclusion. (**B**) Cumulative incidence of MMR in both the TDM arm (continuous line) and the control arm (dashed line) during the 36-month study period. Patients were censored in case of imatinib cessation (as is usual for studies comparing imatinib and second-generation tyrosine kinase inhibitors [6,7,8]). (**C**) The cumulative incidence of MMR in both the TDM arm (continuous line) and the control arm (dashed line) during the 36-month study period. Patients were not censored in case of imatinib cessation.

**Figure 3 pharmaceutics-14-01676-f003:**
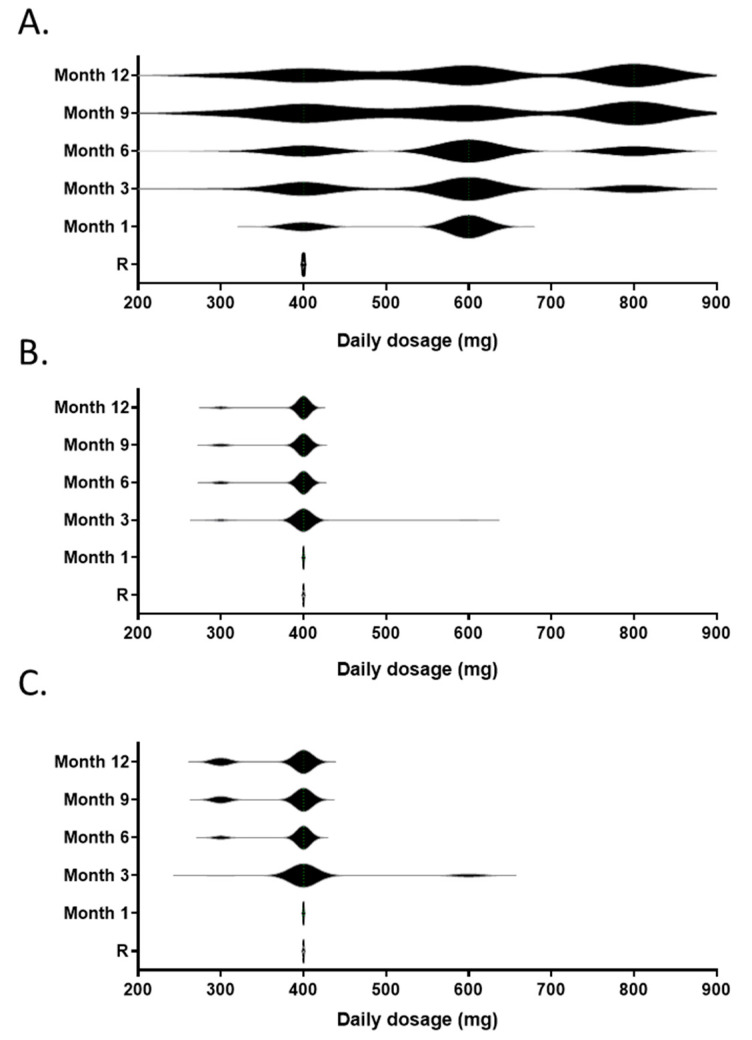
Imatinib daily dose over the 12-month adaptation period for the patients included in the TDM arm (**A**), the control arm (**B**) and the observational arm (**C**). The violin plots represent the distributions of individual daily doses.

**Table 1 pharmaceutics-14-01676-t001:** Characteristics of the patients included in the OPTIM-imatinib trial.

		Initial [C]min <1000 ng/mL		Initial [C]min ≥1000 ng/mL	*p* Value
	Evaluable Patients	TDM Arm	Control Arm	Observational Arm	
Patients (n)	133	43	43	47	
Median age at diagnosis, years (min–max)	64 (27–87)	61 (27–85)		67 (36–87)	0.007
Sex ratio [M/F]	2.09 [90/43]	1.96 [65/33]		1.13 [25/22]	0.17
Median body surface area, m^2^ (min–max)	1.94 (1.09–2.52)	1.96 (1.09–2.44)		1.91 (1.25–2.52)	0.15
Median body weight, kg (min–max)	80 (41–124)	80 (52–124)		78 (41–118)	0.087
Sokal score, n (%) Low/Intermediate/High/NA	37/73/20/3 (28/55/15/2)	24/46/13/3 (28/53/15/4)		13/27/7/0 (28/57/15/0)	0.97
Additional chromosomal abnormalities, n (major route)	5 (1)	3 (1)		2 (none)	0.79
Median time between diagnosis and inclusion, weeks (range)	5 (0–16)	6 (0–16)		3 (0–16)	0.025
Median [C]min at inclusion ng/mL (range)	808 (236–2292)	619 (236–998)		1286 (1002–2292)	<0.0001
Patients with imatinib before inclusion: n, (%)	68 (51)	44 (51)		24 (51)	1.0

**Table 2 pharmaceutics-14-01676-t002:** Pharmacokinetic results during the 12 months follow-up of the OPTIM imatinib study.

Median [C]min ng/mL (95% CI)	Initial Assessment	M3	M6	M9	M12
TDM	602 (546–672)	845 (762–1033)	987 (748–1261)	1088 (860–1219)	971 (830–1242)
Control	651 (558–786)	564 (487–717)	601 (539–722)	541 (422–730)	639 (494–729)
Observational	1286 (1199–1465)	1009 (872–1135)	984 (784–1197)	935 (743–1175)	963 (845–1098)

TDM: therapeutic drug monitoring.

## Data Availability

DRCI, Centre Hospitalier de Versailles.

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
