# Peer review of "Imatinib Optimized Therapy Improves Major Molecular Response Rates in Patients with Chronic Myeloid Leukemia"

_pharmaceutics, 2022, doi:10.3390/pharmaceutics14081676_

Round 1
Reviewer 1 Report
This publication provides evidence for TDM to be used with imatinib treatment, however a significant failure in the paper is that there is no information provided on the method used for the analysis of the imatinib, even the reference provided for the analysis of imatinib does not describe the method of analysis. This is a significant omission of the paper.
Some additional comments include:
Change plasmatic to plasma throughout the manuscript for example Lines 41, 85, 148, 149, 153, 172, 194
Line 106, duplicate of mass remove one
Line 107, 132 reference 23 does not describe the LCMS method,
What was the dose increasing strategy? Was it just increase to 600mg and remeasure and then increase to 800mg and remeasure, this needs to be included
Line 133 plasmatic dosage should read plasma dose
Why p<0.25 line 151
Table 1 line 175, patient characteristics for the control and TDM arm for <1000ng/mL should be separated to see if there are any differences in these 2 arms
Table 1- why are units in nM in the Initial Cmin when ng/mL is used everywhere else
Did the observation arm always have a Cmin >1000ng/mL?
Author Response
We thank the reviewers for their careful review of our manuscript
Reviewer 1.
This publication provides evidence for TDM to be used with imatinib treatment, however a significant failure in the paper is that there is no information provided on the method used for the analysis of the imatinib, even the reference provided for the analysis of imatinib does not describe the method of analysis. This is a significant omission of the paper.
We thank the reviewer for highlighting the mistake in reference 23. We apologize for the error and we corrected reference 23 with the appropriate paper. We also added a brief sentence in the method section to describe the method used for the OPTIM Imatinib study :
“Briefly, after a liquid-liquid extraction, the imatinib and its deuterated internal standard were eluted on an XTerra RP18 column with a gradient of acetonitrile-ammonium formiate buffer 4 mmol/l, pH 3.2. Imatinib was detected by electrospray ionization mass spectrometry with multiple reaction-monitoring mode. The calibration curves were linear over the range 10-5000 ng/ml. The limit of quantification was set at 10 ng/ml.”
Some additional comments include:
Change plasmatic to plasma throughout the manuscript for example Lines 41, 85, 148, 149, 153, 172, 194
We made the requested changes
Line 106, duplicate of mass remove one
We corrected the duplicate, thank you
Line 107, 132 reference 23 does not describe the LCMS method,
We thank the reviewer, the appropriate reference has been inserted in reference 23
What was the dose increasing strategy? Was it just increase to 600mg and remeasure and then increase to 800mg and remeasure, this needs to be included
Yes the dose was increased up to 800 mg per day if the plasma dosage was inadequate at 600 mg. This is mentioned in the text line 150 :
“In the absence of grade ≥ 2 adverse events, patients allocated to the TDM arm were proposed to increase the imatinib daily dose from 400 mg to 600 mg and from 600 mg to 800 mg with the objective to reach the threshold of 1000 ng/ml.”
We have modified the sentence as follow :
“In the absence of grade ≥ 2 adverse events, patients allocated to the TDM arm were proposed to increase the imatinib daily dose from 400 mg to 600 mg and imatinib plasma dosage was remeasured. If the threshold of 1000 ng/ml was not achieved, then patients were proposed to increase again the imatinib daily dose from 600 mg to 800 mg.”
Line 133 plasmatic dosage should read plasma dose
This point was corrected
Why p<0.25 line 151
This value relies to the H0 hypothesis.
Table 1 line 175, patient characteristics for the control and TDM arm for <1000ng/mL should be separated to see if there are any differences in these 2 arms
The randomization was well balanced without differences between the two arms. A sentence has been added.
“The randomization was well balanced and no differences in terms of age, sex ratio, Sokal risk score were observed between patients included in the TDM and control arms.”
We prefer not to add a supplementary column because our point was to highlight the differences between patients with a low Cmin and patients with a high Cmin.
Table 1- why are units in nM in the Initial Cmin when ng/mL is used everywhere else
We thank the reviewer for this remark. This was a spelling error and the heads of the table was corrected in ng/ml.
Did the observation arm always have a Cmin >1000ng/mL?
No and we observed little variations in the Cmin as highlighted in table 2. As reported, the median Cmin in the observation arm tended to decrease with time and the lowest values observed were 1199 ng/ml at randomization, 872 ng/ml at 3 months, 784 ng/ml at 6 months, 743 ng/ml at 9 months and 845 ng/ml at 12 months, perhaps reflecting a dose adaptation for tolerance in these patients (fig 3C).

Reviewer 2 Report
Johnson-Ansah et al. study the value of TDM in patients with chronic phase chronic myelogenous leukemia treated with imatinib in the manuscript. Imatinib is a well analyzed and studied tyrosine kinase inhibitor that is FDA approved. The safety of long-term therapy is well established and investigated.
Nevertheless, even the usage of well-known therapeutics for a long period of time leaves space for new insights. The manuscript “Imatinib optimized therapy improves major molecular response rates in patients with chronic myeloid leukemia” is well written and easy to follow. The authors included 133 patients in their study over 12 months. The results show that a dose optimization for patients diagnosed with CML can be beneficial during their treatment. These findings are very important for the future of treatment of leukemia patients.
The manuscript meets the criteria for publication in the present form. I only suggest to include a legend in figure 2B for better understanding.
Author Response
Reviewer 2.
Johnson-Ansah et al. study the value of TDM in patients with chronic phase chronic myelogenous leukemia treated with imatinib in the manuscript. Imatinib is a well analyzed and studied tyrosine kinase inhibitor that is FDA approved. The safety of long-term therapy is well established and investigated.
Nevertheless, even the usage of well-known therapeutics for a long period of time leaves space for new insights. The manuscript “Imatinib optimized therapy improves major molecular response rates in patients with chronic myeloid leukemia” is well written and easy to follow. The authors included 133 patients in their study over 12 months. The results show that a dose optimization for patients diagnosed with CML can be beneficial during their treatment. These findings are very important for the future of treatment of leukemia patients.
The manuscript meets the criteria for publication in the present form. I only suggest to include a legend in figure 2B for better understanding.
We thank reviewer 2. A legend as been added in figure 2B.
2B. Cumulative incidence of MMR in both the TDM arm (continuous line) and the control arm (dashed line) during the 36 months study period. Patients were censored in case of imatinib cessation as reported by studies comparing imatinib and second generation tyrosine kinase inhibitors [6-8].

Round 2
Reviewer 1 Report
Thankyou to the authors for making the suggested changes.